# Does education sculpt healthcare choices? Exploring factors influencing healthcare utilization among female youths in eight low and lower-middle-income countries

MD Nahid Hassan Nishan[1]*, Khadiza Akter[2], Afroza Sharmin[3‡], Tazmin Akter Tithi[2‡], M. Z. E. M. Naser Uddin Ahmed[1‡]

1 Department of Public Health, North South University, Dhaka, Bangladesh, 2 Department of Nursing, International University of Business Agriculture and Technology, Dhaka, Bangladesh, 3 Department of Nursing, HBH International Nursing College, Dhaka, Bangladesh

☯ These authors contributed equally to this work.
‡ These authors also contributed equally to this work.
* nissan0808@yahoo.com

## Abstracts

### Introduction

Emphasizing the global commitment to universal health coverage, this research addresses geographical variations and challenges faced by young females across low and lower-middle-income countries. Therefore, the objective of this study is to determine the factors that influence the decision-making of young females when it comes to seeking healthcare services in low or lower-middle-income countries.

### Methodology

We examined healthcare usage among female youth across eight countries. We used data from the DHS and employed regression and Chi² tests for our analysis. Our focus was on females aged 15 to 24 and their visits to healthcare facilities. To ensure the validity of our findings, we used weighted sampling to represent the population.

### Results

We had a total sample of 51,298 female youth groups between the ages of 15 and 24 who participated in our study. When it comes to the demographics of the participants, most of those in the 15-19 age group were from Burkina Faso (54.26%), while most of those in the 20-24 age group were from Ghana (50.19%). The impact of education varied across countries; primary education led to increased healthcare utilization in Kenya, Tanzania, and Cambodia, but unexpected trends were observed in Burkina Faso.

### Conclusions

Education significantly influences healthcare utilization, positively impacting both primary and secondary education. Rural residents face challenges accessing timely healthcare.

**Data availability statement:** Data is available from the Demographic and Health Surveys (DHS) program. https://dhsprogram.com

**Funding:** The author(s) received no specific funding for this work.

**Competing interests:** The authors have declared that no competing interests exist.

**Abbreviation:** AOR, Adjusted Odds Ratio; DHS, Demographic and Health Survey; EA, Enumeration Areas; ISE, Innovations in Secondary Education; SDG, Sustainable Development Goal; UNDP, United Nations Development Programme

Geographical challenges, like diseases and limited accessibility, contribute to varied healthcare usage in Kenya and the Philippines. Addressing infrastructure issues is critical, emphasizing education and promoting transparency to enhance healthcare equity.

## Introduction

Healthcare utilization encompasses the degree to which people use the services of the healthcare system [1]. Several factors determine healthcare utilization, which can be broadly categorized into two types: service-related and user-related factors. Service factor refers to the accessibility, availability, and quality of the service. Conversely, user factors include individual characteristics such as age, gender, other demographics, specific health needs, and perceived beliefs regarding their health and the healthcare system [1,2]. Though several factors determine healthcare utilization, the ultimate goal of this healthcare utilization is to protect and promote people's health [1]. More importantly, there is a global desire to ensure healthcare utilization for individuals irrespective of their barriers. This commitment is a crucial target that has been declared in the 2030 Agenda [3]. Specifically, this falls under the Sustainable Development Goal (SDG), and the number three goal emphasizes the achievement of universal health coverage to enhance health promotion and well-being for people of all ages, irrespective of their demographic characteristics [4]. So, different organizations (for example, UNDP) and governments are working progressively to achieve this goal [5]. It is believed that the functional healthcare system is very crucial to reaching the SDG goal and supporting universal health coverage by ensuring healthcare utilization [6]. However, evidence shows that the rate of healthcare utilization varies due to geographical differences, and this rate could mostly vary in low and lower-middle-income countries across different population groups [7–9]. Their financial hardship and the underdeveloped healthcare system pose a significant barrier [6]. Considering this issue, global experts are giving special attention to low and lower-middle-income countries so that they can have better healthcare utilization and ensure quality healthcare [10].

As this progressed, studies showed that the majority focused on adult age group people, children under five years old, and often the elderly. For this, there are several incorporated activities in low and lower-middle-income countries, such as insurance coverage for maternal healthcare utilization and primary healthcare services are introduced [11–13]. Even though there have been improvements over the three decades, children and teenagers in low-income and middle-income countries are still not achieving their full health potential [14]. This is more intense, in low and lower-middle-income countries as this creates a burden in many aspects of utilization of healthcare among young female groups [15]. It is crucial to focus on them as studies have shown they are vulnerable to several adverse health consequences, such as poor reproductive health, chronic diseases, complications from substance abuse, mental health disorders, behavioral complexity, suicidal ideation, and many more [14,16,17]. Additionally, these groups face tremendous challenges in seeking healthcare services, and often, in several cases, it is difficult for the female youth as they face more challenges than adult females while receiving healthcare services [18]. Youth groups in lower-middle-income countries face a range of health challenges that affect different aspects of their well-being. Accidental injuries and road traffic accidents are a cause of death and disability among them. Infectious diseases like HIV, tuberculosis, and respiratory infections still impact adolescents, while the ongoing COVID-19 pandemic adds difficulties to their well-being [19]. Early pregnancy and childbirth environmental risks, overweight issues, and malnutrition add to the complexity of health problems faced by this group [19]. In addition, the long-distance of healthcare makes

it difficult for young females to receive proper healthcare services [20]. So, it can be seen that multiple factors influence the reception of healthcare services among youth groups, and this may not be the same across low- and lower-middle-income countries. This raises the question of which factors mostly play a vital role in utilizing health care among young female groups in those countries. If we can identify those crucial factors that are mostly contributing to healthcare utilization among low and lower-middle-income countries, this finding will help the stakeholders and policymakers to consider more effective implementation and intervention to reach the SDG goal properly, and this poses a great novelty in this study. After a thorough literature review, we identified this research gap and therefore, this study aims to identify the factors influencing the female youth's decision to receive healthcare services in low- and lower-middle-income countries.

## Method and materials

We relied on the secondary dataset from the Demographic and Health Survey (DHS) to conduct this study. The DHS contains data from 108 countries globally [21]. However, we chose 62 countries categorized as low-income and lower-middle-income countries by the World Bank ranking system [22]. Eight of these nations have been chosen because they met our interest criterion: the utilization of healthcare facilities among youth groups. The countries enlisted for this study are in table 1.

We followed cross-sectional study techniques using data extracted from the most recent DHS dataset from 2020 and onwards. Although we began searching datasets from 2017, none of the earlier datasets contained all the variables necessary for our study. This DHS survey usually follows a two-step sampling approach to get the data from a desired country. Firstly, it selects the household in-cluster enumeration areas (EAs), and then a defined number of households are chosen from each cluster. The DHS collaborates and empowers the national implementing agencies through regional workshops so that they can ensure the training and other necessary requirements for the field staff to collect data effectively from the household. For detailed information regarding their study site, population, data collection, and sampling process, you may visit [31]. We analyzed this secondary data, focusing on healthcare utilization among young women. The DHS Program is authorized to distribute, at no cost, unrestricted survey data files for legitimate academic research [31].

For this study, we considered the individual recorded information (IR files) obtained from the DHS as it has access to data of females aged 15 to 49 years. Our goal was to get information regarding whether the young generation (female) has utilized health facilities in recent years, so we took the variable "visited a health facility in the last 12 months". This variable captures a broader time frame, reducing potential biases associated with short-term

**Table 1. List of countries used in this study.**

| Country | Year |
|---|---|
| Burkina Faso | 2021 [23] |
| Côte d'Ivoire | 2021 [24] |
| Ghana | 2022 [25] |
| Kenya | 2022 [26] |
| Tanzania | 2022 [27] |
| Cambodia | 2021-2022 [28] |
| Nepal | 2022 [29] |
| Philippines | 2022 [30] |

health-seeking behavior fluctuations` This approach is supported by previous studies that have demonstrated the importance of 12 months for assessing healthcare utilization [32–34]. We restricted it to those above 24 years. We also ensure that there is no presence of missing values. Moreover, any unwanted variables have also been removed. Additionally, weight has been applied to the sample of all countries to adjust them according to the population sizes and survey years.

## Dependent and independent variables

The outcome variable we have used for this study is whether the respondent visited a healthcare facility in the last twelve months. This variable was selected to provide a comprehensive view of healthcare utilization, supported by previous research indicating the effectiveness of 12 months in capturing healthcare-seeking behavior. Depending upon several condition there could be many reasons an individual will visit the healthcare settings [35]. However, for this current study, we only considered a particular group (youth) who received healthcare services through healthcare visits. Moreover, we kept the outcome variable in a binary approach with values 1 and 0 for analysis based on the study objectives. Here, value "1" refers to visited healthcare facilities in the last twelve months, and "0" refers otherwise.

Initially, a total of 10 variables were incorporated as independent variables for this current study due to their access to DHS in these seven countries. However, within these 10 variables, we have modified it to seven explanatory variables to align with our research interest. These variables encompass age, residence, education, wealth index, personal access to vehicle, reaching time to nearest health care facility, and mass media engagement. The primary explanatory variable was the level of education. For our analysis, we have categorized these variables in a specific way. For example, an ordinal scale had been employed for the age variable: "15-19 Age group" and "20-24 Age group". The education variable was categorized as the individual has No education, Primary, Secondary, and Higher. The resident variable was categorized as Urban and Rural.

Furthermore, the wealth index was categorized as Poor, Middle, and Rich. Personal access to a vehicle was determined by combining three variables (household having a bicycle, household having motorcycle/scooter, household having car/truck) and then categorizing them as "Yes"- have access and "No" have no access. Additionally, reaching time to the nearest healthcare facility was categorized as "Within 1 hour", "2-3 Hours," and "More than 3 hours". Finally, mass media engagement had been categorized as "Present" media engagement" and "Absent" media engagement by merging three variables (frequency of reading newspapers or magazines, frequency of watching TV, and frequency of listening to radio). All these covariates were kept based on their significance in any of the countries studied, with a minimum significance value of less than 0.05. Variables that were found significant during bivariate analysis in any of our study countries were retained and used in the final model analysis.

## Data analysis

To analyze our data thoroughly, we relied on Stata, a statistical software version 17. As DHS has a two-stage cluster sampling design, there is a potential for unequal sampling probability, clustering, and stratification. However, we applied the weighting technique through survey (Svy) commands to resolve this issue. The weighting technique enables the fitting of a proper statistical model by ensuring the representativeness of the survey data, and DHS highly recommends this technique [36]. We did not pool data from all countries. Instead, we conducted separate analyses for each country and displayed the results in a table. The same variables and code were used for each country's analysis to ensure consistency.

We begin our analysis by calculating descriptive statistics to present a comprehensive overview of the variables. Afterward, we did the bivariate analysis. We checked the association through the Chi$^2$ test, which helped us to assess the explanatory variables' significance level. We used the ($^{\Psi}$) symbol to highlight the Chi$^2$ significant strengths. We individually examined all explanatory variables to assess their association with the outcome variable and found that all of the independent variables we chose were suitable for multivariate analysis as their *P*-value were < 0.05. Moving forward to multivariate analysis, we applied logistic regression, carefully observed the *P*-value, and presented the significance with the adjusted odd ratio (AOR). To show multivariate AOR strength, we used (*), where more stars represent more strength. We considered a 95% confidence interval for this multivariate analysis. In addition, with these statistical analyses, we have also checked multicollinearity and confirmed there was no presence of multicollinearity, and the model was fitted properly for our analysis. To test for multicollinearity, we created a dummy variable calculated residuals by subtracting predicted values from observed values and generated a correlation matrix among the independent variables and residuals, using a cut point of 0.5. The residuals were not correlated with any independent variables, confirming that multicollinearity was not a significant issue. Additionally, we performed the Hosmer-Lemeshow goodness-of-fit test with the group value set to 10, and the *P*-value exceeded 0.5, indicating a good fit [37]. Lastly, we constructed tables to present our study findings.

We have included S1 Fig, a flowchart that demonstrates the entire data selection to the analysis process, to provide a clear and visual explanation of the variables and the methodology used in our study.

## Results

We got a total of 51,298 weighted youth aged 15 to 24 participants in our study. In terms of demographic characteristics, most participant's 15-19-year age group were in Burkina Faso (54.26%), and most participants from the 20-24 years group were from Ghana (50.19%). The majority of the participants from Nepal live in urban areas (68.09%), and most Burkina Faso participants are from rural areas (64.84%). About (43.90%) received primary education, whereas (68.86%) and (27.90%) received secondary and higher education in the Philippines. More information is in Table 2.

In a bivariate analysis Table 3 shows, there is a highly significant association (*p*-value < 0.001) present between the age group and visiting healthcare to receive healthcare facilities, and this significance is present in all of the countries of our study. Moreover, the residents have a strong association with outcomes in Cambodia and the Philippines (*p*-value < 0.001), but Côte d'Ivoire showed a weak statistical association (*p*-value < 0.05). Furthermore, education showed a high statistically significant association (*p*-value < 0.001) in multiple countries (Burkina Faso, Côte d'Ivoire, Ghana, Kenya, Cambodia, and the Philippines). Additionally, the wealth index and media engagement also showed a high statistically significant association (*p*-value < 0.001) in Kenya, Cambodia, and the Philippines, but a moderate association (*p*-value < 0.01) was only found for the wealth index and a weak association (*p*-value < 0.05) for media engagement in Burkina Faso. Other variables include access to personal transport, having a moderate association (*p*-value < 0.01) in Tanzania and Nepal, and reaching time to nearby healthcare facilities, which only showed a weak association (*p*-value < 0.05) in Nepal.

### Demographic factor

In our study Table 4 shows, age plays a significant role in healthcare utilization among youth. Individuals aged 20-24 years across Burkina Faso, Côte d'Ivoire, Ghana, Kenya, Tanzania,

Cambodia, Nepal, and the Philippines showed a higher likelihood of accessing healthcare services compared to the 15-19 years age group. The adjusted odds ratios are respectively: (AOR: 3.44 CI:3.04-3.89), (AOR: 2.75 CI: 2.39-3.17), (AOR: 2.41 CI: 2.08-2.76), (AOR: 2.46 CI: 2.11-2.87), (AOR: 2.93, CI: 2.55-3.37), (AOR: 2.25, CI: 1.91-2.64), (AOR: 2.64, CI: 2.32-3.01), and (AOR: 3.87, CI: 3.22-4.64) and all of them are highly statistically significant.

The important key factor, education, and its influence on healthcare utilization are evident in Burkina Faso, Côte d'Ivoire, Kenya, Tanzania, and Cambodia. In Kenya, Tanzania, and Cambodia, individuals with primary education showed a higher likelihood (AOR 2.07 CI:1.50-2.85), (AOR 1.36 CI:1.09-1.68), and (AOR 1.55 CI:1.01-2.37) of visiting healthcare facilities compared to those without education except Burkina Faso where there is a lower likelihood in primary education (AOR 0.84 CI:0.72-0.98) compared to those without education, with statistical significance. Additionally, Burkina Faso also showed a lower likelihood (AOR 0.83 CI:0.73-0.95) of secondary education compared to those who are not educated, but Côte d'Ivoire, Kenya, and Tanzania showed the opposite, where they showed a higher likelihood of healthcare utilization (AOR 1.31 CI:1.07-1.57), (AOR 2.58 CI:1.88-3.54) and (AOR 1.42 CI:1.11-1.80), respectively. Only Kenya showed a higher likelihood of (AOR 2.16 CI:1.48-3.17) healthcare utilization among higher educated individuals compared to those who were not educated, and this is also highly significant.

**Table 2. Demographic characteristics of individuals.**

| | Sub-Saharan Africa Region | | | | | Southeast Asian Region | | |
| --- | --- | --- | --- | --- | --- | --- | --- | --- |
| | Burkina Faso | Côte d'Ivoire | Ghana | Kenya | Tanzania | Cambodia | Nepal | Philippine |
| Characteristics | N (%) | N (%) | N (%) | N (%) | N (%) | N (%) | N (%) | N (%) |
| Youth Age group | | | | | | | | |
| 15-19 year | 3846 (54.26) | 3176 (54.15) | 2672 (49.81) | 3122 (50.52) | 3078 (53.05) | 2966 (53.42) | 2637 (50.06) | 5513 (54.16) |
| 20-24 year | 3242(45.74) | 2690 (45.85) | 2692 (50.19) | 3057 (49.48) | 2724 (46.95) | 2586 (46.58) | 2631 (49.94) | 4666 (45.84) |
| Resident | | | | | | | | |
| Urban | 2492 (35.16) | 3867 (65.91) | 3002 (55.97) | 2424 (39.23) | 2072 (35.71) | 2261 (40.72) | 3587 (68.09) | 5492 (53.95) |
| Rural | 4596 (64.84) | 1999 (34.09) | 2362 (44.03) | 3755 (60.77) | 3730 (64.29) | 3291 (59.28) | 1681 (31.91) | 4687 (46.05) |
| Wealth Index | | | | | | | | |
| Middle | 1304 (18.40) | 1189 (20.28) | 1221 (22.77) | 1145 (18.51) | 1107 (19.09) | 1129 (20.34) | 1058 (20.07) | 2117 (20.80) |
| Poor | 2377 (33.53) | 1765 (30.09) | 1967 (36.67) | 2265 (36.65) | 1917 (33.04) | 2026 (36.50) | 2086 (39.61) | 3627 (35.63) |
| Rich | 3407 (48.07) | 2912 (49.63) | 2176 (40.56) | 2769 (44.81) | 2778 (47.87) | 2397 (43.16) | 2124 (40.32) | 4435 (43.57) |
| Education | | | | | | | | |
| No Education | 2548 (35.93) | 2001 (34.12) | 292 (5.46) | 160 (2.60) | 635 (10.95) | 151 (2.73) | 362 (6.88) | 33 (0.33) |
| Primary | 1298 (18.31) | 959 (16.35) | 651 (12.14) | 1590 (25.74) | 2547 (43.90) | 1197 (21.57) | 1673 (31.77) | 296 (2.91) |
| Secondary | 3106 (43.83) | 2644 (45.08) | 4108 (76.58) | 3434 (55.56) | 2581 (44.46) | 3732 (67.19) | 3123 (59.28) | 7099 (68.86) |
| Higher | 136 (1.93) | 262 (4.45) | 313 (5.82) | 995 (16.10) | 39 (0.69) | 472 (8.51) | 110 (2.08) | 2841 (27.90) |
| Has Access to Personal Transport | | | | | | | | |
| No | 1093 (15.43) | 4448 (75.84) | 4089 (76.24) | 5064 (81.95) | 4029 (69.43) | 2609 (47.00) | 3156 (59.92) | 7256 (71.28) |
| Yes | 5995 (84.57) | 1418 (24.16) | 1275 (23.76) | 1115 (18.05) | 1773 (30.57) | 2943 (53.00) | 2112 (40.08) | 2923 (28.72) |
| Media Engagement | | | | | | | | |
| No | 2075 (29.27) | 1499 (25.57) | 1177 (21.93) | 1550 (25.08) | 2329 (40.15) | 2135 (38.46) | 2229 (42.31) | 1298 (12.76) |
| Yes | 5013 (70.73) | 1417 (24.16) | 4187 (78.07) | 4629 (74.92) | 3473 (59.85) | 3417 (61.54) | 3039 (57.69) | 8881 (87.24) |
| Nearby Healthcare Reaching time | | | | | | | | |
| Within 1 hour | 6245 (88.11) | 5623 (95.86) | 5154 (96.07) | 5801 (93.88) | 5167 (89.04) | 5499 (99.03) | 5085 (96.52) | 10118 (99.39) |
| Within 2-3 Hour | 633 (8.93) | 227 (3.87) | 201 (3.76) | 346 (5.60) | 578 (9.97) | 46 (0.84) | 174 (3.31) | 52 (0.51) |
| More than 3 Hour | 210 (2.96) | 16 (0.27) | 9 (0.17) | 32 (0.52) | 57 (0.99) | 7 (0.13) | 9 (0.17) | 9 (0.09) |

**Table 3. Association between influencing factors and healthcare utilization.**

| Characteristics | Sub-Saharan Africa Region | | | | | | | | | | Southeast Asia | | | | | |
|---|---|---|---|---|---|---|---|---|---|---|---|---|---|---|---|---|
| | Burkina Faso | | Côte d'Ivoire | | Ghana | | Kenya | | Tanzania | | Cambodia | | Nepal | | Philippine | |
| | No (N%) | Yes (N%) | No (N%) | Yes (N%) | No (N%) | Yes (N%) | No (N%) | Yes (N%) | No (N%) | Yes (N%) | No (N%) | Yes (N%) | No (N%) | Yes (N%) | No (N%) | Yes (N%) |
| Youth Age group | p=0.000[ΨΨΨ] | | p=0.000[ΨΨΨ] | | p=0.000[ΨΨΨ] | | p=0.000[ΨΨΨ] | | p=0.000[ΨΨΨ] | | p=0.000[ΨΨΨ] | | p=0.000[ΨΨΨ] | | p=0.000[ΨΨΨ] | |
| 15-19 year | 2281 (70.63) | 1564 (40.55) | 2234 (63.82) | 941 (39.83) | 1845 (58.82) | 826 (37.12) | 1955 (60.60) | 1167 (39.52) | 2068 (64.48) | 1010 (38.93) | 2338 (58.90) | 628 (39.68) | 1202 (65.53) | 1434 (41.78) | 4883 (58.41) | 630 (34.63) |
| 20-24 year | 948 (29.37) | 2293 (59.45) | 1266 (36.18) | 1422 (60.17) | 1292 (41.18) | 1399 (62.88) | 1271 (39.40) | 1786 (60.48) | 1139 (35.52) | 1585 (61.07) | 1631 (41.10) | 955 (60.32) | 632 (34.47) | 1998 (58.22) | 3477 (41.59) | 1189 (65.37) |
| Resident | p=0.749 | | p=0.018[Ψ] | | p=0.475 | | p=0.677 | | p=0.881 | | p=0.000[ΨΨΨ] | | p=0.371 | | p=0.000[ΨΨΨ] | |
| Urban | 1148 (35.57) | 1343 (34.82) | 2345 (66.98) | 1520 (64.31) | 1784 (56.87) | 1217 (54.71) | 1241 (38.48) | 1182 (40.05) | 1132 (35.30) | 940 (36.22) | 1705 (42.96) | 555 (35.10) | 1227 (66.88) | 2359 (68.74) | 4627 (55.34) | 865 (47.57) |
| Rural | 2080 (64.43) | 2514 (65.18) | 1155 (33.02) | 843 (35.69) | 1353 (43.13) | 1008 (45.29) | 1985 (61.52) | 1770 (59.95) | 2075 (64.70) | 1655 (63.78) | 2264 (57.04) | 1027 (64.90) | 607 (33.12) | 1073 (31.26) | 3733 (44.66) | 953 (52.43) |
| Wealth Index | p=0.002[ΨΨ] | | p=0.544 | | p=0.683 | | p=0.000[ΨΨΨ] | | p=0.135 | | p=0.000[ΨΨΨ] | | p=0.958 | | p=0.000[ΨΨΨ] | |
| Middle | 566 (17.53) | 738 (19.13) | 732 (20.92) | 457 (19.34) | 702 (22.39) | 518 (23.31) | 603 (18.71) | 541 (18.35) | 627 (19.57) | 480 (18.50) | 794 (20.02) | 334 (21.15) | 362 (19.77) | 694 (20.24) | 1732 (20.72) | 384 (21.13) |
| Poor | 1137 (35.23) | 1238 (32.10) | 1041 (29.75) | 723 (30.60) | 1161 (36.99) | 806 (36.22) | 1254 (38.89) | 1010 (34.21) | 1053 (32.85) | 864 (33.28) | 1326 (33.42) | 699 (44.20) | 743 (40.53) | 1343 (39.12) | 2878 (34.43) | 748 (41.16) |
| Rich | 1525 (47.24) | 1881 (48.76) | 1727 (49.33) | 1183 (50.07) | 1274 (40.61) | 900 (40.47) | 1368 (42.41) | 1401 (47.44) | 1526 (47.58) | 1251 (48.22) | 1847 (46.55) | 548 (34.66) | 728 (39.70) | 1395 (40.64) | 3749 (44.85) | 686 (37.71) |
| Education | p=0.000[ΨΨΨ] | | p=0.000[ΨΨΨ] | | p=0.000[ΨΨΨ] | | p=0.000[ΨΨΨ] | | p=0.188 | | p=0.000[ΨΨΨ] | | p=0.113 | | p=0.000[ΨΨΨ] | |
| No Education | 1029 (31.87) | 1517 (39.32) | 1209 (34.55) | 791 (33.49) | 156 (4.97) | 136 (6.13) | 112 (3.48) | 48.24 (1.63) | 368 (11.50) | 266 (10.28) | 105 (2.65) | 46 (2.93) | 117 (6.38) | 245 (7.15) | 29 (0.35) | 4 (0.26) |
| Primary | 645 (19.99) | 652 (16.90) | 545 (15.57) | 414 (17.52) | 403 (12.87) | 247 (11.11) | 909 (28.18) | 681 (23.07) | 1366 (42.62) | 1180 (45.48) | 778 (19.61) | 419 (26.47) | 584 (31.83) | 1089 (31.74) | 206 (2.46) | 90 (4.98) |
| Secondary | 1507 (46.67) | 1599 (41.46) | 1623 (46.37) | 1020 (43.17) | 2428 (77.37) | 1679 (75.47) | 1752 (54.31) | 1681 (56.93) | 1457 (45.45) | 1122 (43.23) | 2725 (68.67) | 1005 (63.47) | 1106 (60.29) | 81 (2.38) | 5814 (69.55) | 1194 (65.68) |
| Higher | 47 (1.47) | 89 (2.32) | 123 (3.52) | 137 (5.83) | 150 (4.78) | 162 (7.29) | 452 (14.04) | 542 (18.36) | 13 (0.43) | 26 (1.01) | 359 (9.07) | 112 (7.13) | 27 (1.50) | 0 (0.00) | 2310 (27.64) | 529 (29.08) |
| Has Access to Personal Transport | p=0.066 | | p=0.431 | | p=0.314 | | p=0.443 | | p=0.005[ΨΨ] | | p=0.000[ΨΨΨ] | | p=0.007[ΨΨ] | | p=0.179 | |
| No | 463 (14.34) | 630 (16.34) | 2648 (75.63) | 1800 (76.14) | 2389 (76.15) | 1699 (76.36) | 2637 (81.73) | 2427 (82.18) | 2170 (67.66) | 1859 (71.62) | 1729 (43.56) | 880 (55.63) | 1054 (57.43) | 2102 (61.260) | 5957 (71.25) | 1298 (71.39) |
| Yes | 2766 (85.66) | 3228 (83.66) | 853 (24.37) | 564 (23.86) | 748 (23.85) | 526 (23.64) | 589 (18.27) | 526 (17.82) | 1037 (32.34) | 736 (28.38) | 2240 (56.44) | 702 (44.37) | 781 (42.57) | 1330 (38.74) | 2403 (28.75) | 520 (28.61) |
| Media Engagement | p=0.033[Ψ] | | p=0.279 | | p=0.304 | | p=0.000[ΨΨΨ] | | p=0.298 | | p=0.000[ΨΨΨ] | | p=0.270 | | p=0.000[ΨΨΨ] | |
| No | 903 (27.97) | 1171 (30.36) | 885 (25.30) | 613 (25.96) | 692 (22.07) | 484 (21.75) | 845 (26.21) | 704 (23.84) | 1301 (40.58) | 1028 (39.62) | 1483 (37.37) | 652 (41.21) | 756 (41.20) | 1473 (42.91) | 1007 (291) | 291 (16.02) |
| Yes | 2326 (72.03) | 2687 (69.64) | 2615 (74.70) | 1750 (74.04) | 2445 (77.93) | 1741 (78.25) | 2380 (73.79) | 2249 (76.16) | 1905 (59.42) | 1567 (60.38) | 2486 (62.63) | 930 (58.79) | 1079 (58.80) | 1960 (57.09) | 7353 (87.95) | 1527 (83.98) |
| Nearby Healthcare Reaching time | p=0.796 | | p=0.110 | | p=0.147 | | p=0.056 | | p=0.616 | | p=0.106 | | p=0.028[Ψ] | | p=0.742 | |
| Within 1 hour | 2845 (88.10) | 3400 (88.12) | 3350 (95.70) | 2272 (96.10) | 3006 (95.81) | 2146 (96.43) | 3012 (93.34) | 2790 (94.47) | 2857 (89.08) | 2310 (88.99) | 3938 (99.23) | 1560 (98.53) | 1769 (96.43) | 3315 (96.57) | 8306 (99.35) | 1811 (99.59) |
| Within 2-3 Hour | 290 (8.99) | 342 (8.88) | 136 (3.91) | 90 (3.81) | 128 (4.10) | 73 (0.28) | 194 (6.04) | 150 (5.11) | 310 (9.68) | 268 (10.34) | 25 (0.64) | 21 (1.34) | 59 (3.22) | 115 (3.36) | 46 (0.55) | 6 (0.33) |
| More than 3 Hour | 94 (2.92) | 115 (3.00) | 13 (0.39) | 2 (0.10) | 3 (0.09) | 6.20 (0.28) | 19 (0.62) | 12 (0.42) | 39 (1.24) | 17 (0.67) | 5 (0.13) | 2 (0.33) | 6 (0.35) | 2 (0.07) | 8 (0.10) | 1 (0.08) |

Denote: Chi² significance: [Ψ]= *P*-value<0.05, [ΨΨ]= *P*-value<0.01, [ΨΨΨ]= *P*-value<0.001)

**Table 4. Influencing factors of healthcare utilization among female youth groups in low and lower-middle-income countries.**

| | Sub-Saharan Africa Region | | | | | | | | | | Southeast Asia | | | | | |
|---|---|---|---|---|---|---|---|---|---|---|---|---|---|---|---|---|
| | Burkina Faso | | Côte d'Ivoire | | Ghana | | Kenya | | Tanzania | | Cambodia | | Nepal | | Philippine | |
| Characteristics | AOR | CI | AOR | CI | AOR | CI | AOR | CI | AOR | CI | AOR | CI | AOR | CI | AOR | CI |
| Youth Age group | | | | | | | | | | | | | | | | |
| 15-19 year | Ref | | Ref | | Ref | | Ref | | Ref | | Ref | | Ref | | Ref | |
| 20-24 year | 3.44 | 3.04-3.89*** | 2.75 | 2.39-3.17*** | 2.41 | 2.08-2.76*** | 2.46 | 2.11-2.87*** | 2.93 | 2.55-3.37*** | 2.25 | 1.91-2.64*** | 2.64 | 2.32-3.01*** | 3.87 | 3.22-4.64*** |
| Resident | | | | | | | | | | | | | | | | |
| Urban | Ref | | Ref | | Ref | | Ref | | Ref | | Ref | | Ref | | Ref | |
| Rural | 1.18 | 0.99-1.40 | 1.17 | 0.98-1.38 | 1.18 | 0.99-1.39 | 1.24 | 1.00-1.53* | 0.99 | 0.79-1.23 | 1.14 | 0.94-1.41 | 0.93 | 0.80-1.08 | 1.41 | 1.17-1.70*** |
| Wealth Index | | | | | | | | | | | | | | | | |
| Middle | Ref | | Ref | | Ref | | Ref | | Ref | | Ref | | Ref | | Ref | |
| Poor | 0.73 | 0.62-0.87*** | 1.07 | 0.88-1.31 | 0.91 | 0.74-1.10 | 0.97 | 0.80-1.18 | 1.11 | 0.90-1.37 | 1.21 | 0.99-1.49 | 0.91 | 0.75-1.11 | 1.03 | 0.83-1.27 |
| Rich | 1.01 | 0.84-1.19 | 1.14 | 0.93-1.39 | 0.92 | 0.75-1.12 | 1.13 | 0.91-1.41 | 0.93 | 0.75-1.16 | 0.73 | 0.59-0.92** | 0.96 | 0.78-1.18 | 0.91 | 0.74-1.12 |
| Education | | | | | | | | | | | | | | | | |
| No Education | Ref | | Ref | | Ref | | Ref | | Ref | | Ref | | Ref | | Ref | |
| Primary | 0.84 | 0.72-0.98* | 1.22 | 0.93-1.60 | 0.88 | 0.64-1.21 | 2.07 | 1.50-2.85*** | 1.36 | 1.09-1.68** | 1.55 | 1.01-2.37* | 1.05 | 0.75-1.47 | 3.46 | 0.99-12.05 |
| Secondary | 0.83 | 0.73-0.95* | 1.31 | 1.07-1.57** | 0.92 | 0.70-1.20 | 2.58 | 1.88-3.54*** | 1.42 | 1.11-1.80** | 1.41 | 0.92-2.10 | 0.96 | 0.68-1.35 | 2.32 | 0.70-7.67 |
| Higher | 0.79 | 0.51-1.24 | 1.37 | 0.92-2.06 | 1.05 | 0.66-1.66 | 2.16 | 1.48-3.17*** | 1.91 | 0.88-4.07 | 0.97 | 0.58-1.62 | 0.92 | 0.46-1.87 | 1.16 | 0.34-3.94 |
| Has Access to Personal Transport | | | | | | | | | | | | | | | | |
| No | Ref | | Ref | | Ref | | Ref | | Ref | | Ref | | Ref | | Ref | |
| Yes | 0.89 | 0.74-1.06 | 0.99 | 0.84-1.18 | 0.99 | 0.83-1.18 | 0.99 | 0.82-1.20 | 0.84 | 0.72-0.99* | 0.71 | 0.59-0.82*** | 0.85 | 0.73-1.01 | 1.11 | 0.91-1.31 |
| Media Engagement | | | | | | | | | | | | | | | | |
| No | Ref | | Ref | | Ref | | Ref | | Ref | | Ref | | Ref | | Ref | |
| Yes | 0.86 | 0.75-1.01 | 0.95 | 0.79-1.15 | 1.06 | 0.90-1.26 | 1.02 | 0.87-1.20 | 1.14 | 0.98-1.33 | 1.11 | 0.93-1.31 | 0.94 | 0.80-1.10 | 0.83 | 0.65-1.05 |
| Nearby Healthcare Reaching time | | | | | | | | | | | | | | | | |
| Within 1 hour | Ref | | Ref | | Ref | | Ref | | Ref | | Ref | | Ref | | Ref | |
| Within 2-3 Hour | 0.96 | 0.77-1.19 | 0.88 | 0.63-1.25 | 0.77 | 0.53-1.12 | 0.98 | 0.75-1.29 | 1.11 | 0.86-1.43 | 1.43 | 0.77-2.65 | 0.97 | 0.71-1.31 | 0.49 | 0.23-1.06 |
| More than 3 Hour | 1.01 | 0.65-1.59 | 0.27 | 0.11-0.67** | 3.13 | 0.90-10.77 | 0.94 | 0.52-1.72 | 0.51 | 0.20-1.27 | 1.01 | 0.25-3.99 | 0.18 | 0.02-1.18 | 0.81 | 0.20-3.26 |

Denote: *P*-value indication: *= *P*-value<0.05, **= *P*-value<0.01, ***= *P*-value<0.001, AOR=adjusted odds ratio)

However, in terms of residents, there is an increased likelihood of healthcare utilization seen in rural individuals in the Sub-Saharan African region: Kenya, and in the Southeast Asian region: Philippines with (AOR 1.41 CI: 1.17-1.70) and (AOR 1.24 CI: 1.00-1.53) compared to the urban people.

## Socio-economic factor

In the wealth index context, Burkina Faso and Cambodia showed a significant role. In Burkina Faso, the poor have a lower likelihood (AOR 0.73 CI:0.62-0.87), and in Cambodia rich also have a lower likelihood (AOR 0.73 CI:0.59-0.92) of healthcare utilization compared to the middle class, and both are significant.

In Tanzania and Cambodia, those who have personal transport both have a lower likelihood (AOR 0.84 CI:0.72-0.99) and (AOR 0.71 CI:0.59-0.82) of healthcare utilization compared to those who do not, and this result is also statistically significant.

## Healthcare accessibility

Table 5 shows a significant association across all the regions of our study between Urban-rural settings and Nearby healthcare reaching time. There is a noticeable trend across all of the regions, which is that there is a higher percentage of individuals who live in urban regions have reached within 1 hour compared to those who live in rural areas.

On the other hand, multivariate analysis showed in Côte d'Ivoire that those who require more than 3 hours to reach nearby health facilities have a lower likelihood (AOR 0.27 CI:0.11-0.62) of healthcare utilization compared to those who have within 1-hour reaching time, and this is moderately statistically significant.

## Discussions

The comprehensive exploration of healthcare utilization among youth reveals intricate dynamics shaped by socio-demographic contexts, with significant variations across countries. For a better understanding of our results, we have discussed them using a multi-angular approach, which ultimately contributes to more effective and equitable healthcare systems. This approach is supported in contemporary health services research literature, demonstrating the value of integrating statistical analysis with theoretical frameworks, highlighting practical implications for high-quality health systems, and employing comparative analysis to examine health inequalities across regions and such extensive examinations validate the effectiveness of our approach [38–41].

Our analysis reveals significant associations between urban-rural settings and healthcare access times. Urban residents more frequently reach healthcare facilities within an hour compared to rural residents, likely due to better infrastructure and transportation systems.

**Table 5. Association between residence and time required to reach nearby healthcare facilities.**

| | Sub-Saharan Africa Region | | | | | | | | | | | | | | Southeast Asia | | | | | |
| | Burkina Faso | | Côte d'Ivoire | | Ghana | | Kenya | | Tanzania | | Cambodia | | Nepal | | Philippine | |
| Reaching Time Nearest Healthcare | p=0.000ΨΨΨ | | p=0.000ΨΨΨ | | p=0.000ΨΨΨ | | p=0.000ΨΨΨ | | p=0.000ΨΨΨ | | p=0.000ΨΨΨ | | p=0.000ΨΨΨ | | p=0.002ΨΨ | |
| | Urban | Rural | Urban | Rural | Urban | Rural | Urban | Rural | Urban | Rural | Urban | Rural | Urban | Rural | Urban | Rural |
| | N (%) | N (%) | N (%) | N (%) | N (%) | N (%) | N (%) | N (%) | N (%) | N (%) | N (%) | N (%) | N (%) | N (%) | N (%) | N (%) |
| 1 hour | 2424 (97.29) | 3820 (83.13) | 3776 (97.69) | 1846 (92.34) | 2954 (98.40) | 2199 (93.11) | 2370 (97.76) | 3432 (91.38) | 2021 (97.53) | 3146 (84.32) | 2256 (99.81) | 3242 (98.50) | 3501 (97.62) | 1583 (94.17) | 5471 (99.62) | 4646 (99.13) |
| 2-3 hours | 22 (0.91) | 610 (13.28) | 81 (2.10) | 145 (7.29) | 46 (1.53) | 155 (6.59) | 53 (2.21) | 292 (7.78) | 46 (2.22) | 532 (14.28) | 2 (0.08) | 44 (1.36) | 81 (2.28) | 92 (5.53) | 15 (0.28) | 37 (0.79) |
| More than 3 hours | 44 (1.80) | 164 (3.59) | 8 (0.21) | 7 (0.38) | 2 (0.07) | 6 (0.29) | 1 (0.04) | 31 (0.84) | 5 (0.25) | 52 (1.40) | 2 (0.11) | 4 (0.14) | 3 (0.11) | 4 (0.30) | 5 (0.11) | 3 (0.08) |

Denote: Chi² significance: Ψ= *P*-value<0.05, ΨΨ= *P*-value<0.01, ΨΨΨ= *P*-value<0.001)

This trend, observed consistently across all regions, emphasizes disparities in healthcare access due to infrastructural differences. Additionally, the 20-24 age group accesses healthcare more frequently than the 15-19 age group, attributed to increased health awareness, emerging health needs, and greater autonomy in decision-making typically seen in early adulthood [42]. However, regional variations indicate that the one-size-fits-all approach may not be effective. In Côte d'Ivoire and Burkina Faso, long travel times significantly hinder healthcare utilization, suggesting there is a need for mobile clinics and transport subsidies which may improve their healthcare access. Meanwhile, in the Philippines, where geographical barriers are pronounced, telemedicine services and decentralized healthcare facilities can bridge the accessibility gap. Existing models like Muso's Pro-CCM model in Mali, which deploys community health workers for home-based care, could be adapted to improve early healthcare engagement in these settings [43–45].

The study findings align with established theories on health behavior and social determinants of health. Urban areas' better infrastructure and higher concentration of healthcare facilities support the idea that improved access leads to higher healthcare utilization. The age-related differences reflect developmental perspectives that emphasize increased health consciousness and autonomy during early adulthood. Education's impact on healthcare utilization in countries like Kenya, Tanzania, and Cambodia supports the notion that education enhances health literacy and promotes proactive health behaviors [46]. Basic education helps develop health awareness, empowering individuals to prioritize well-being and seek appropriate healthcare services. Health literacy, encompassing skills like accessing, understanding, evaluating, and applying health information, is often enhanced through primary education, enabling informed health decisions [47]. To make education more effective for healthcare utilization, country-specific strategies like, in Kenya and Tanzania, adding health literacy to school curricula could reinforce early health habits. In Burkina Faso, where formal education alone isn't enough, community-led programs like Jeunes en Vigie, which trains young women to evaluate health services, could be more impactful [48]The results underscore the need for targeted interventions to improve healthcare access in rural areas. Enhancing rural infrastructure, such as roads and transportation systems, can reduce travel times to healthcare facilities. Health education programs targeting adolescents, particularly those in the 15-19 age group, can address barriers related to lack of awareness and dependence on family members for healthcare decisions. In Burkina Faso, Côte d'Ivoire, Ghana, Kenya, Tanzania, Cambodia, Nepal, and the Philippines, the 20-24 age group shows a higher likelihood of accessing healthcare services compared to the 15-19 age group, attributed to increased health awareness, autonomy in decision-making, and improved access to resources during early adulthood [49–51].

Policies that increase access to education can indirectly improve healthcare utilization by enhancing health literacy and empowering individuals to seek necessary medical services. In Kenya, Tanzania, and Cambodia, individuals with primary education show a higher likelihood of healthcare utilization compared to those without education. This suggests that education fosters health awareness and knowledge [46]. Secondary education further promotes health-seeking behavior, as seen in countries like Côte d'Ivoire, Kenya, and Tanzania. The Innovations in Secondary Education (ISE) initiative, promoting secondary education and empowering youth, could significantly increase healthcare utilization [52]. Research indicates that higher education correlates with better health and longer lifespans, motivating individuals to seek healthcare services [53]. However, in Burkina Faso, those with primary or secondary education show lower healthcare utilization, possibly due to cultural beliefs or self-care practices [54].

The study's findings highlight regional contrasts and support the need for tailored solutions. While Kenya, Tanzania, and Cambodia show that primary education improves

healthcare utilization, Burkina Faso presents an inverse pattern, suggesting that one-size-fits-all solutions may not be effective. Rural individuals in Kenya and the Philippines exhibit higher healthcare utilization due to higher disease prevalence, contrasting with lower utilization in rural areas of other countries due to accessibility issues. Residency shows an increased likelihood of healthcare utilization in rural individuals in Kenya and the Philippines compared to urban areas. In Kenya, remote regions' prevalence of infectious diseases and other conditions may burden rural populations, triggering higher healthcare utilization [55].

The archipelagic nature of the Philippines presents obstacles to healthcare accessibility in isolated rural areas [56]. In rural settings, many may live in distinct mountain areas where healthcare access is not easy. Furthermore, the higher frequency of diseases and lack of access to pharmacies or shops for self-medication may lead to more frequent healthcare visits among female youth. In Burkina Faso, poor individuals show a lower likelihood of healthcare utilization compared to the middle class, possibly due to financial barriers. Since young women often do not earn enough, they may face financial issues limiting their healthcare access [57,58]. In Cambodia, the wealthy may show lower healthcare utilization due to better overall health and low trust in the healthcare system [59]. In Tanzania and Cambodia, lower healthcare utilization among those with personal transportation may result from dependence on family members for travel to distant healthcare facilities [39]. To address this, cash-transfer programs that incentivize preventive healthcare visits and subsidized health insurance for young women could help. In Cambodia, improving service quality and transparency could rebuild confidence in wealthier people. Kenya's Zuri Health platform, which provides telemedicine through mobile phones, could also serve as a model to expand healthcare access in rural areas [60].

In Côte d'Ivoire, individuals requiring more than 3 hours to reach nearby health facilities exhibit a lower likelihood of healthcare utilization compared to those within a 1-hour reach. The diverse terrain and lack of sufficient healthcare facilities pose obstacles to timely access. Challenges with transportation, such as poorly maintained roads or lack of convenient public transportation, further complicate access [61]. To ensure sustainable progress, governments and organizations should implement nationwide health equity monitoring systems that track disparities in real-time, measure policy impact, and allow for evidence-based adjustments. This would help address evolving challenges and refine interventions to fit local needs.

## Limitations

The study's reliance on self-reported data which may introduce subjective biases. The temporal scope is limited to post-2020 data. The omission of countries due to insufficient data affects generalizability as the representation of eight countries may not capture broader regional nuances. Uneven sample sizes across regions may introduce biases in precision and representativeness. Lack of exploration into participants' cultural backgrounds misses insights into cultural influences on healthcare utilization patterns. These limitations suggest areas for future research to enhance understanding.

## Conclusions

This research shows that education plays a significant role in healthcare usage across the countries, with primary and secondary education positively influencing it. The connection between residence and the time it takes to get to healthcare facilities highlights difficulties among rural individuals when it comes to accessing services promptly. It's fascinating that as young adults transition into adulthood between the ages of 20 and 24, they tend to become more conscious about their health, embrace their independence, and gain access to resources.

However, Burkina Faso presents a contrast where educated individuals may prioritize self-care over seeking medical attention. Additionally, educated youth females may face barriers due to gender and cultural beliefs in Burkina Faso, which result in lower healthcare utilization. Geography, diseases, and limited accessibility may further contribute to increased utilization of healthcare services in areas in Kenya and the Philippines. The situation in Côte d'Ivoire highlights an urgent need for intervention in the poorly maintained roads and convenient public transportation. Considering these findings, it is recommended that we focus on initiatives to enhance healthcare facilities in regions, overcome cultural and gender-related obstacles among female youth, and foster trust in health systems by promoting transparency and accountability. It is also important to focus on promoting female education and health literacy in different regions. Initiatives such as the UNESCO strategy on education for health and well-being should address education to empower females. Policymakers and healthcare professionals should work together to guarantee that health resources are distributed equitably and minimize the obstacles to healthcare facilities among youth.

## Supporting information

**S1 Fig.** Flowchart of data selection and analysis process.
(DOCX)

## Acknowledgments

We thank the Demographics and Health Survey (DHS) for providing us with access to their survey data.

## Author contributions

**Conceptualization:** MD Nahid Hassan Nishan.

**Data curation:** MD Nahid Hassan Nishan.

**Formal analysis:** MD Nahid Hassan Nishan.

**Methodology:** MD Nahid Hassan Nishan, Khadiza Akter, Afroza Sharmin.

**Software:** MD Nahid Hassan Nishan.

**Visualization:** MD Nahid Hassan Nishan.

**Writing – original draft:** MD Nahid Hassan Nishan, Khadiza Akter.

**Writing – review & editing:** MD Nahid Hassan Nishan, Khadiza Akter, Afroza Sharmin, Tazmin Akter Tithi, M Z E M Naser Uddin Ahmed.

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
