## [Decision Letter · Decision Letter 0]

23 Jun 2024

PONE-D-24-04110Does Education Sculpt Healthcare Choices? Exploring Factors Influencing Healthcare Utilization Among Female Youths in Eight Low and Lower-Middle-Income CountriesPLOS ONE

Dear Dr. NISHAN,

Thank you for submitting your manuscript to PLOS ONE. After careful consideration, we feel that it has merit but does not fully meet PLOS ONE’s publication criteria as it currently stands. Therefore, we invite you to submit a revised version of the manuscript that addresses the points raised during the review process.

We look forward to receiving your revised manuscript.

Kind regards,

Innocent Besigye

Academic Editor

PLOS ONE

Reviewers' comments:

Reviewer's Responses to Questions

**Comments to the Author**

1. Is the manuscript technically sound, and do the data support the conclusions?

Reviewer #1: Partly

Reviewer #2: Yes

2. Has the statistical analysis been performed appropriately and rigorously? 

Reviewer #1: No

Reviewer #2: Yes

3. Have the authors made all data underlying the findings in their manuscript fully available?

Reviewer #1: Yes

Reviewer #2: Yes

4. Is the manuscript presented in an intelligible fashion and written in standard English?

Reviewer #1: No

Reviewer #2: No

5. Review Comments to the Author

Reviewer #1: The article touches an interesting and useful point in the context of the SDGs, tracking their progress and inequalities in access across SSA.

I was surprised to see the choice of the outcome variable as a proxy for access to health overall. I would suggest the authors to research a bit more on what variables are usually used, from the DHS surveys, to analyse access to health facilities. The one chosen could lead to potential biases as it only reports accesses in the past month. Please explore other options and/or cite previous published studies where the same variable has been used for similar purposes.

Statistical analyses:

-No covariate selection process done and arbitrary choice of covariates for the model. Usually, from an initial literature review where a list of covariates are pre-selected based on literature, statistical techniques are used to select covariates that go into the final model.

-The choice of covariates should be driven by the statistical selection and not from the authors.

-Testing for multicollinearity is mentioned in the text, however no details were given (e.g. what tests were used / what cut off was considered). Testing for interactions between covariates has not been performed or reported.

-Not clear from the text if the authors have pooled the data for all the countries together or not. In the case of a pooled dataset, I believe the recommendation from the DHS Program is to transform the weights (from normalised to de-normalised) before undertaking any analysis. This is because the sampling weights are Country specific, and therefore not comparable. I would strongly recommend the authors to consult the DHS material on this matter, and to re-run the data if using a pooled dataset.

-Not clear if exactly the same covariates were used for all the countries - if this is a country specific study then separate covariate selection processes should be performed.

There is no justification as to why pooling all the Countries together in the study, especially if results at the end are given at Country Level. Please expand on this.

More specifically, regarding the regression analysis method chosen, here are some considerations that the authors may want to consider:

If I understand well the authors have used the svyset command in STATA to weight the data and take into account, the sampling structure and included the weighting (see above for more about sampling weights). Authors have cited the DHS blog (ref 33) however the link does not work (at least on my hand).

If working with a pooled dataset and having as a goal to be able to have country- based outputs, I would suggest the authors to explore the multilevel modelling techniques which are often used with the DHS data. Multilevel modelling techniques deal with the complex and clustered sampling data structure, and the hierarchical nature of the DHS data. Since multiple observations (women - level 1) are nested within clusters and multiple countries are nested withing the same country, multilevel modelling deals with hierarchical structure of the data and the within and between cluster correlations of the observations, and therefore lead to more reliable results, while controlling for intra cluster correlation.

The use of multilevel modelling though would need careful consideration, including a thorough check of the minimum number of units needed at each level and the construction of the group weights. Please explore previous studies and some books on the topic such as (just to give a few examples): Harvey Goldstein, Multilevel Statistical Modelling; Applied Multilevel Analysis: A Practical Guide. Jos WR Twisk. and the material from the Centre for Multilevel Modelling from the University of Bristol. Also, see a recent publication from the DHS Program on MULTILEVEL MODELING USING DHS SURVEYS: A FRAMEWORK TO APPROXIMATE LEVEL-WEIGHTS – DHS METHODOLOGICAL REPORTS 27

English Language

Language used is at bits too colloquial and sometimes there are mistakes (e.g. Line 152: The resident has kept a dichotomy: Urban and Rural.) Please revise the English, as the manuscript is not always up to the standards of an academic journal article.

Reviewer #2: • It is well known that education plays a significant role in healthcare usage across countries, and education positively influences it. Many studies mention that like (Rizvi DS. Health education and global health: Practices, applications, and future research. J Educ Health Promot. 2022 Aug 25;11:262. doi: 10.4103/jehp.jehp_218_22. PMID: 36325224; PMCID: PMC9621358). What did this study add to what was already known on this topic?

• The manuscript needs English editing

• The title is informative but so long. It is better to be considered

• Line 51: it is better to write ‘’Education significantly influences healthcare utilization, positively impacting both primary and secondary education’’

• The aim is clear

• What is the role of the researchers? Is that collected data from the Demographic and Health Survey (DHS). You did not collect data by yourself

• Are you sure the variables are defined and measured appropriately?

• Why did you not use a flowchart to explain the variables

• The results need to be discussed from multiple angles and placed into context without being overinterpreted

6. PLOS authors have the option to publish the peer review history of their article (what does this mean? ). If published, this will include your full peer review and any attached files.

**Do you want your identity to be public for this peer review?** For information about this choice, including consent withdrawal, please see our Privacy Policy .

Reviewer #1: No

Reviewer #2: **Yes: ** Dr. Enas Fakhry Abdel Hamed

---

## [Author Response · Author response to Decision Letter 1]

29 Jun 2024

Dear Reviewers,

Thank you for your insightful and constructive feedback on our manuscript. We have carefully considered your comments and made the following revisions:

Reviewer 1:

Comment 1: The article touches on an interesting and useful point in the context of the SDGs, tracking their progress and inequalities in access across SSA. I was surprised to see the choice of the outcome variable as a proxy for access to health overall. I would suggest the authors to research a bit more on what variables are usually used, from the DHS surveys, to analyze access to health facilities. The one chosen could lead to potential biases as it only reports accesses in the past month. Please explore other options and/or cite previous published studies where the same variable has been used for similar purposes.

Response 1: Thank you for your insightful comment. However, we used the "Visited health facility in the last 12 months" variable, not just visits in the past month, as our outcome variable. This choice captures a broader time frame, reducing potential bias associated with short-term health-seeking behavior fluctuations. Similar variables have been successfully used in previous studies. For example, a study using the Medical Expenditure Panel Survey found that 56.3% of children aged 0 to 18 years had no well-child visits during a 12-month period, highlighting the importance of this time frame in capturing health-seeking behavior. Additionally, research focusing on long-term care services emphasized the significance of considering longer periods when assessing healthcare utilization, particularly among the aging population. Furthermore, studies examining trends in healthcare utilization beyond immediate periods, such as during lockdowns, reinforce the need for extended time frames to understand comprehensive healthcare-seeking behaviors. Therefore, we may say these examples support the appropriateness of using a 12-month period to evaluate healthcare utilization. We have also, added such a section in lines (137, and 146) and cited the necessary paper inside the manuscript.

Comment 2: No covariate selection process was done and an arbitrary choice of covariates for the model. Usually, from an initial literature review where a list of covariates is pre-selected based on literature, statistical techniques are used to select covariates that go into the final model.

-The choice of covariates should be driven by the statistical selection and not from the authors.

Response 2: We appreciate your concern. The selection of covariates was not arbitrary. Through literature review, we identified potential covariates and then we included covariates that showed a significant result in any of the countries we studied, with a minimum significance value of less than 0.05. If any variable was significant in any country we have chosen, it was retained and used in the final model. We have now additionally included this in the manuscript at (line 170). Thanks.

Comment 3: Testing for multicollinearity is mentioned in the text, however, no details were given (e.g. what tests were used / what cut-off was considered). Testing for interactions between covariates has not been performed or reported.

Response 3: Thank you for pointing this out. To address multicollinearity, we conducted a thorough diagnostic process. Given that we used the “SVY” command, we created a dummy variable to calculate the residuals by subtracting the predicted values from the observed values. We then generated a correlation matrix to examine relationships among the independent variables and the residuals, using a cut point of 0.5 to determine significant multicollinearity. The results indicated that the residuals were not correlated with any of the independent variables, confirming that multicollinearity was not a significant issue. Additionally, we performed the Hosmer-Lemeshow goodness-of-fit test with the group value set to 10, and the p-value exceeded 0.5, indicating a good fit. We have also added this to the manuscript (line 194) to avoid getting further comments.

Comment 4: Not clear from the text if the authors have pooled the data for all the countries together or not. In the case of a pooled dataset, I believe the recommendation from the DHS Program is to transform the weights (from normalized to de-normalized) before undertaking any analysis. This is because the sampling weights are country-specific, and therefore not comparable. I would strongly recommend the authors consult the DHS material on this matter and re-run the data if using a pooled dataset.

Response 4: We appreciate your concern for clarification. We did not combine data from all countries at once. Instead, we conducted separate analyses for each country and displayed the results in a table. The same variables and code were used for each country's analysis to ensure consistency. We have added this information, in the manuscript section at (line 180).

Comment 5: If I understand well the authors have used the svyset command in STATA to weight the data and take into account, the sampling structure, and included the weighting (see above for more about sampling weights). The authors have cited the DHS blog (ref 33) however the link does not work (at least on my hand). If working with a pooled dataset and having as a goal to be able to have country-based outputs, I would suggest the authors to explore the multilevel modelling techniques which are often used with the DHS data. Multilevel modelling techniques deal with the complex and clustered sampling data structure, and the hierarchical nature of the DHS data. Since multiple observations (women - level 1) are nested within clusters and multiple countries are nested within the same country, multilevel modelling deals with hierarchical structure of the data and the within and between cluster correlations of the observations, and therefore lead to more reliable results, while controlling for intra cluster correlation. The use of multilevel modelling though would need careful consideration, including a thorough check of the minimum number of units needed at each level and the construction of the group weights. Please explore previous studies and some books on the topic such as (just to give a few examples): Harvey Goldstein, Multilevel Statistical Modelling; Applied Multilevel Analysis: A Practical Guide. Jos WR Twisk. and the material from the Centre for Multilevel Modelling at the University of Bristol. Also, see a recent publication from the DHS Program on multilevel modeling using DHS surveys: a framework to approximate level-weights – DHS methodological reports 27.

Response 5: Thank you for your suggestion. Since we did not use a pooled dataset, multilevel modelling techniques was not required for our analysis. Each country's data was analyzed individually, and the results were displayed separately. Moreover, regarding the Ref (33) issue (new Ref 31), we have checked the URL again and it seems is it working fine. Please turn off any VPN or any other firewall that might block the Browser from opening such URLs. Thanks for your suggestion.

Comment 6: The language used is at bits too colloquial and sometimes there are mistakes (e.g. Line 152: The resident has kept a dichotomy: Urban and Rural.) Please revise the English, as the manuscript is not always up to the standards of an academic journal article.

Response 6: We appreciate your feedback on language usage. We have reviewed the entire manuscript again, edited it for language, and corrected any grammatical mistakes to meet academic standards. Thanks a lot for the feedback.

Reviewer 2:

Comment 1: It is well known that education plays a significant role in healthcare usage across countries, and education positively influences it. Many studies mention that like (Rizvi DS. Health education and global health: Practices, applications, and future research. J Educ Health Promote. 2022 Aug 25; 11:262. Doi: 10.4103/jehp.jehp_218_22. PMID: 36325224; PMCID: PMC9621358). What did this study add to what was already known on this topic?

Response 1: We appreciate your insightful comment. As we all know, education significantly influences healthcare usage, but there can be variations across different regions. Although this has been demonstrated in many studies, including "Health Education and Global Health: Practices, Applications, and Future Research," our study is unique in that it examines multiple regions simultaneously and identifies diverse patterns in each. No previous study has analyzed these countries together to draw such comprehensive conclusions. Additionally, we only included data from the DHS Program collected after 2017, with all our data from 2021 and onward, providing the most up-to-date information for this analysis.

Comment 2: The manuscript needs English editing.

Response 2: We appreciate your feedback on English usage. We have reviewed the entire manuscript again and edited it for language and clarity to meet academic standards. Thanks a lot for the feedback.

Comment 3: The title is informative but so long. It is better to be considered.

Response 3: We understand your concern about the title length. However, we believe it is essential to provide clear and comprehensive information to readers. The title aims to convey the study's scope and focus, ensuring that potential readers understand the study's relevance and context. We appreciate your understanding in maintaining the title's current form to achieve this clarity.

Comment 4: Line 51: it is better to write ‘’Education significantly influences healthcare utilization, positively impacting both primary and secondary education’’

Response 4: Thanks for addressing this. The line now changed in (line 51).

Comment 5: The aim is clear.

Response 5: Thanks for clarifying this.

Comment 6: What is the role of the researchers? Is that collected data from the Demographic and Health Survey (DHS)? You did not collect data by yourself?

Response 6: Thank you for highlighting the need for clarity regarding data collection. The data for this study was obtained from the Demographic and Health Surveys (DHS) Program, which collects data through standardized surveys in various countries. The researchers analyzed this secondary data, focusing on healthcare utilization among women. The DHS Program is authorized to distribute, at no cost, unrestricted survey data files for legitimate academic research (https://dhsprogram.com/data/data-collection.cfm). This information is now also added in (line 131) in the main manuscript.

Comment 7: Are you sure the variables are defined and measured appropriately? Why did you not use a flowchart to explain the variables?

Response 7: Thank you for this important point. We have rechecked everything and we ensure all variables are measured properly. For better understanding, an additional supplementary flowchart has been added in a diagram manner to make it comprehensive to understand the procedure we did. Thanks for your concern. This is cited in (line 201) and the supplementary file has been linked at (line 550).

Comment 8: The results need to be discussed from multiple angles and placed into context without being overinterpreted.

Response 8: We appreciate your suggestion and we precise the discussion section to demonstrate it from various perspectives and provide a broader context, ensuring a comprehensive understanding of the findings. A significant change has been made in the discussion section of the manuscript. Thanks for your consideration.

---

## [Editor Report · Decision Letter 1]

9 Jul 2024

PONE-D-24-04110R1

Does Education Sculpt Healthcare Choices? Exploring Factors Influencing Healthcare Utilization Among Female Youths in Eight Low and Lower-Middle-Income Countries

PLOS ONE

Dear Dr. NISHAN,

Thank you for submitting your manuscript to PLOS ONE. After careful consideration, we have decided that your manuscript does not meet our criteria for publication and must therefore be rejected.

I am sorry that we cannot be more positive on this occasion, but hope that you appreciate the reasons for this decision.

Kind regards,

Innocent Besigye

Academic Editor

PLOS ONE

Additional Editor Comments:

Dear author(s),

Thank you for attempting to respond to the reviewers. However, most of the responses to the reviewers' comments are counter-arguments instead of appropriately responding strengthen the scientific and scholarly merits of the manuscript.

- - - - -

---

## [Author Response · Author response to Decision Letter 2]

17 Jul 2024

Dear Reviewers,

Thank you for your insightful and constructive feedback on our manuscript. We have carefully considered your comments and made the following revisions:

Reviewer 1:

Comment 1: The article touches on an interesting and useful point in the context of the SDGs, tracking their progress and inequalities in access across SSA. I was surprised to see the choice of the outcome variable as a proxy for access to health overall. I would suggest the authors to research a bit more on what variables are usually used, from the DHS surveys, to analyze access to health facilities. The one chosen could lead to potential biases as it only reports accesses in the past month. Please explore other options and/or cite previous published studies where the same variable has been used for similar purposes.

Response 1: Thank you for your insightful comment. However, we used the "Visited health facility in the last 12 months" variable, not just visits in the past month, as our outcome variable. This choice captures a broader time frame, reducing potential bias associated with short-term health-seeking behavior fluctuations. Similar variables have been successfully used in previous studies. For example, a study using the Medical Expenditure Panel Survey found that 56.3% of children aged 0 to 18 years had no well-child visits during a 12-month period, highlighting the importance of this time frame in capturing health-seeking behavior. Additionally, research focusing on long-term care services emphasized the significance of considering longer periods when assessing healthcare utilization, particularly among the aging population. Furthermore, studies examining trends in healthcare utilization beyond immediate periods, such as during lockdowns, reinforce the need for extended time frames to understand comprehensive healthcare-seeking behaviors. Therefore, we may say these examples support the appropriateness of using a 12-month period to evaluate healthcare utilization. We have also, added such a section in lines (137, and 146) and cited the necessary paper inside the manuscript.

Comment 2: No covariate selection process was done and an arbitrary choice of covariates for the model. Usually, from an initial literature review where a list of covariates is pre-selected based on literature, statistical techniques are used to select covariates that go into the final model.

-The choice of covariates should be driven by the statistical selection and not from the authors.

Response 2: We appreciate your concern. The selection of covariates was not arbitrary. Through literature review, we identified potential covariates and then we included covariates that showed a significant result in any of the countries we studied, with a minimum significance value of less than 0.05. If any variable was significant in any country we have chosen, it was retained and used in the final model. We have now additionally included this in the manuscript at (line 170). Thanks.

Comment 3: Testing for multicollinearity is mentioned in the text, however, no details were given (e.g. what tests were used / what cut-off was considered). Testing for interactions between covariates has not been performed or reported.

Response 3: Thank you for pointing this out. To address multicollinearity, we conducted a thorough diagnostic process. Given that we used the “SVY” command, we created a dummy variable to calculate the residuals by subtracting the predicted values from the observed values. We then generated a correlation matrix to examine relationships among the independent variables and the residuals, using a cut point of 0.5 to determine significant multicollinearity. The results indicated that the residuals were not correlated with any of the independent variables, confirming that multicollinearity was not a significant issue. Additionally, we performed the Hosmer-Lemeshow goodness-of-fit test with the group value set to 10, and the p-value exceeded 0.5, indicating a good fit. We have also added this to the manuscript (line 194) to avoid getting further comments.

Comment 4: Not clear from the text if the authors have pooled the data for all the countries together or not. In the case of a pooled dataset, I believe the recommendation from the DHS Program is to transform the weights (from normalized to de-normalized) before undertaking any analysis. This is because the sampling weights are country-specific, and therefore not comparable. I would strongly recommend the authors consult the DHS material on this matter and re-run the data if using a pooled dataset.

Response 4: We appreciate your concern for clarification. We did not combine data from all countries at once. Instead, we conducted separate analyses for each country and displayed the results in a table. The same variables and code were used for each country's analysis to ensure consistency. We have added this information, in the manuscript section at (line 180).

Comment 5: If I understand well the authors have used the svyset command in STATA to weight the data and take into account, the sampling structure, and included the weighting (see above for more about sampling weights). The authors have cited the DHS blog (ref 33) however the link does not work (at least on my hand). If working with a pooled dataset and having as a goal to be able to have country-based outputs, I would suggest the authors to explore the multilevel modelling techniques which are often used with the DHS data. Multilevel modelling techniques deal with the complex and clustered sampling data structure, and the hierarchical nature of the DHS data. Since multiple observations (women - level 1) are nested within clusters and multiple countries are nested within the same country, multilevel modelling deals with hierarchical structure of the data and the within and between cluster correlations of the observations, and therefore lead to more reliable results, while controlling for intra cluster correlation. The use of multilevel modelling though would need careful consideration, including a thorough check of the minimum number of units needed at each level and the construction of the group weights. Please explore previous studies and some books on the topic such as (just to give a few examples): Harvey Goldstein, Multilevel Statistical Modelling; Applied Multilevel Analysis: A Practical Guide. Jos WR Twisk. and the material from the Centre for Multilevel Modelling at the University of Bristol. Also, see a recent publication from the DHS Program on multilevel modeling using DHS surveys: a framework to approximate level-weights – DHS methodological reports 27.

Response 5: Thank you for your suggestion. Since we did not use a pooled dataset, multilevel modelling techniques was not required for our analysis. Each country's data was analyzed individually, and the results were displayed separately. Moreover, regarding the Ref (33) issue (new Ref 31), we have checked the URL again and it seems is it working fine. Please turn off any VPN or any other firewall that might block the Browser from opening such URLs. Thanks for your suggestion.

Comment 6: The language used is at bits too colloquial and sometimes there are mistakes (e.g. Line 152: The resident has kept a dichotomy: Urban and Rural.) Please revise the English, as the manuscript is not always up to the standards of an academic journal article.

Response 6: We appreciate your feedback on language usage. We have reviewed the entire manuscript again, edited it for language, and corrected any grammatical mistakes to meet academic standards. Thanks a lot for the feedback.

Reviewer 2:

Comment 1: It is well known that education plays a significant role in healthcare usage across countries, and education positively influences it. Many studies mention that like (Rizvi DS. Health education and global health: Practices, applications, and future research. J Educ Health Promote. 2022 Aug 25; 11:262. Doi: 10.4103/jehp.jehp_218_22. PMID: 36325224; PMCID: PMC9621358). What did this study add to what was already known on this topic?

Response 1: We appreciate your insightful comment. As we all know, education significantly influences healthcare usage, but there can be variations across different regions. Although this has been demonstrated in many studies, including "Health Education and Global Health: Practices, Applications, and Future Research," our study is unique in that it examines multiple regions simultaneously and identifies diverse patterns in each. No previous study has analyzed these countries together to draw such comprehensive conclusions. Additionally, we only included data from the DHS Program collected after 2017, with all our data from 2021 and onward, providing the most up-to-date information for this analysis.

Comment 2: The manuscript needs English editing.

Response 2: We appreciate your feedback on English usage. We have reviewed the entire manuscript again and edited it for language and clarity to meet academic standards. Thanks a lot for the feedback.

Comment 3: The title is informative but so long. It is better to be considered.

Response 3: We understand your concern about the title length. However, we believe it is essential to provide clear and comprehensive information to readers. The title aims to convey the study's scope and focus, ensuring that potential readers understand the study's relevance and context. We appreciate your understanding in maintaining the title's current form to achieve this clarity.

Comment 4: Line 51: it is better to write ‘’Education significantly influences healthcare utilization, positively impacting both primary and secondary education’’

Response 4: Thanks for addressing this. The line now changed in (line 51).

Comment 5: The aim is clear.

Response 5: Thanks for clarifying this.

Comment 6: What is the role of the researchers? Is that collected data from the Demographic and Health Survey (DHS)? You did not collect data by yourself?

Response 6: Thank you for highlighting the need for clarity regarding data collection. The data for this study was obtained from the Demographic and Health Surveys (DHS) Program, which collects data through standardized surveys in various countries. The researchers analyzed this secondary data, focusing on healthcare utilization among women. The DHS Program is authorized to distribute, at no cost, unrestricted survey data files for legitimate academic research (https://dhsprogram.com/data/data-collection.cfm). This information is now also added in (line 131) in the main manuscript.

Comment 7: Are you sure the variables are defined and measured appropriately? Why did you not use a flowchart to explain the variables?

Response 7: Thank you for this important point. We have rechecked everything and we ensure all variables are measured properly. For better understanding, an additional supplementary flowchart has been added in a diagram manner to make it comprehensive to understand the procedure we did. Thanks for your concern. This is cited in (line 201) and the supplementary file has been linked at (line 550).

Comment 8: The results need to be discussed from multiple angles and placed into context without being overinterpreted.

Response 8: We appreciate your suggestion and we precise the discussion section to demonstrate it from various perspectives and provide a broader context, ensuring a comprehensive understanding of the findings. A significant change has been made in the discussion section of the manuscript. Thanks for your consideration.

---

## [Decision Letter · Decision Letter 2]

7 Feb 2025

PONE-D-24-04110R2Does Education Sculpt Healthcare Choices? Exploring Factors Influencing Healthcare Utilization Among Female Youths in Eight Low and Lower-Middle-Income CountriesPLOS ONE

Dear Dr. NISHAN,

Thank you for submitting your manuscript to PLOS ONE. After careful consideration, we feel that it has merit but does not fully meet PLOS ONE’s publication criteria as it currently stands. Therefore, we invite you to submit a revised version of the manuscript that addresses the points raised during the review process.

**ACADEMIC EDITOR:**

Dear Authors,

Thank you for your patience and for trusting in Plos One for sending your manuscript.

The responses to the reviewers are satisfactory and the modifications made in the text have improved the manuscript. Below we send one last modification, necessary to allow the publication of this study once this necessary action by the researchers has been completed.

Minor Revision:

The discussion of results could benefit from a deeper analysis, particularly regarding the observed regional variations. For instance, the implications of this manuscript could be more explicit about the concrete policy consequences of the results. While the need to improve infrastructure and promote education is mentioned, more specific recommendations could be provided on how governments and organizations can address inequities in healthcare access. This could include proposals for specific programs tailored to each regional context or strategies to enhance the monitoring of certain public policies. It would be valuable to explore in greater detail the cultural, socioeconomic, or other factors that might have implications in these policies.

We look forward to receiving your revised manuscript.

Kind regards,

Javier Fagundo-Rivera, PhD

Academic Editor

PLOS ONE

Journal Requirements:

**Academic Editor Comments:**

Dear Authors,

Thank you for your patience and for trusting in Plos One for sending your manuscript.

The responses to the reviewers are satisfactory and the modifications made in the text have improved the manuscript. Below we send one last modification necessary to allow the publication of this study, once this necessary action by the researchers has been completed.

Minor Revision:

The discussion of results could benefit from a deeper analysis, particularly regarding the observed regional variations. For instance, the implications of this manuscript could be more explicit about the concrete policy consequences of the results. While the need to improve infrastructure and promote education is mentioned, more specific recommendations could be provided on how governments and organizations can address inequities in healthcare access. This could include proposals for specific programs tailored to each regional context or strategies to enhance the monitoring of certain public policies. It would be valuable to explore in greater detail the cultural, socioeconomic, or other factors that might have implications in these policies.

Reviewers' comments:

Reviewer's Responses to Questions

**Comments to the Author**

1. If the authors have adequately addressed your comments raised in a previous round of review and you feel that this manuscript is now acceptable for publication, you may indicate that here to bypass the “Comments to the Author” section, enter your conflict of interest statement in the “Confidential to Editor” section, and submit your "Accept" recommendation.

Reviewer #2: (No Response)

2. Is the manuscript technically sound, and do the data support the conclusions?

Reviewer #2: Partly

3. Has the statistical analysis been performed appropriately and rigorously? 

Reviewer #2: Yes

4. Have the authors made all data underlying the findings in their manuscript fully available?

Reviewer #2: Yes

5. Is the manuscript presented in an intelligible fashion and written in standard English?

Reviewer #2: Yes

6. Review Comments to the Author

Reviewer #2: (No Response)

7. PLOS authors have the option to publish the peer review history of their article (what does this mean? ). If published, this will include your full peer review and any attached files.

**Do you want your identity to be public for this peer review?** For information about this choice, including consent withdrawal, please see our Privacy Policy .

Reviewer #2: **Yes: ** Dr.Enas Abdel Hamed

---

## [Author Response · Author response to Decision Letter 3]

9 Feb 2025

Academic editor comment

Comment 1: The discussion of results could benefit from a deeper analysis, particularly regarding the observed regional variations. For instance, the implications of this manuscript could be more explicit about the concrete policy consequences of the results. While the need to improve infrastructure and promote education is mentioned, more specific recommendations could be provided on how governments and organizations can address inequities in healthcare access. This could include proposals for specific programs tailored to each regional context or strategies to enhance the monitoring of certain public policies. It would be valuable to explore in greater detail the cultural, socioeconomic, or other factors that might have implications in these policies.

Response 1: Thanks. We have revised the manuscript and cited necessary references accordingly. Thank you for your suggestion.

---

## [Editor Report · Decision Letter 3]

12 Feb 2025

Does Education Sculpt Healthcare Choices? Exploring Factors Influencing Healthcare Utilization Among Female Youths in Eight Low and Lower-Middle-Income Countries

PONE-D-24-04110R3

Dear Dr. NISHAN,

We’re pleased to inform you that your manuscript has been judged scientifically suitable for publication and will be formally accepted for publication once it meets all outstanding technical requirements.

Kind regards,

Javier Fagundo-Rivera, PhD

Academic Editor

PLOS ONE

Additional Editor Comments:

Dear authors, we sincerely appreciate your patience and the continuous revisions made to this manuscript. The changes introduced have improved the text, and it is now worthy of publication. Congratulations on its final acceptance.

---

## [Editor Report · Acceptance letter]

PONE-D-24-04110R3

PLOS ONE

Dear Dr. NISHAN,

I'm pleased to inform you that your manuscript has been deemed suitable for publication in PLOS ONE. Congratulations! Your manuscript is now being handed over to our production team.

Kind regards,

on behalf of

Dr. Javier Fagundo-Rivera

Academic Editor

PLOS ONE